# Glycine Betaine-Mediated Root Priming Improves Water Stress Tolerance in Wheat (*Triticum aestivum* L.)

Nazir Ahmed [1,2], Mingyuan Zhu [1], Qiuxia Li [1], Xilei Wang [1], Jiachi Wan [1] and Yushi Zhang [1,*]

[1] Engineering Research Center of Plant Growth Regulator, Ministry of Education, College of Agronomy and Biotechnology, China Agricultural University, Beijing 100193, China; nachachar@sau.edu.pk (N.A.); zhumingyuan@cau.edu.cn (M.Z.); liqiuxia@cau.edu.cn (Q.L.); xileiwang@cau.edu.cn (X.W.); wanjiachi@cau.edu.cn (J.W.)

[2] Department of Crop Physiology, Faculty of Crop Production, Sindh Agriculture University, Tandojam 70060, Pakistan

* Correspondence: zhangyushi@cau.edu.cn

**Abstract:** Droughts represent one of the main challenges that climate change imposes on crop production. As a globally cultivated staple crop, wheat (*Triticum aestivum* L.) is prone to drought environments. Therefore, improvement in drought tolerance represents a growing concern to ensure food security, especially for wheat. In this perspective, the application of Phyto-phillic exogenous materials such as glycine-betaine (GB) has been attracting attention, particularly in stress-related studies. Since roots procure the water and nutrients for plants, any improvements in their response and capacity against drought stress could induce stress tolerance in plants. However, the knowledge about the changes in root architecture, defense mechanism, hormonal metabolism, and downstream signaling, in response to GB-mediated root priming, is still limited. Therefore, we designed the present study to investigate the role of GB-mediated root priming in improving the water stress tolerance in wheat (cv. Jimai-22) under in-vitro conditions. The roots of twelve days old wheat seedlings were treated with Hoagland's solution (GB-0), 50 mM GB (GB-1), and 100 mM GB (GB-2) for 48 h and subjected to well-watered (WW) and water-stress (WS) conditions. The osmotic stress substantially impaired shoot/root growth, dry matter accumulation, and increased malondialdehyde (MDA) and hydrogen-peroxide ($H_2O_2$) production in the roots of wheat seedlings. However, GB-mediated root priming improved the redox homeostasis of wheat roots by boosting the activities of SOD and POD and triggering the significantly higher accumulation of abscisic acid (ABA) and salicylic acid (SA) in the roots of GB-primed plants. Consequently, it modified the root architecture system and improved plant growth, dry matter accumulation, and water-stress tolerance of wheat seedlings. Moreover, GB-mediated root priming increased root sensitivity to water stress and induced overexpression of stress-responsive genes involved in ABA metabolism (*TaNECD1*, *TaABA'OH2*), their downstream signal transduction (*TaPP2C*, *TaSNRK2.8*), and activation of different transcriptional factors (*TabZIP60*, *TaAREB3*, *TaWRKY2*, *TaERF3*, and *TaMYB3*) that are associated with plant metabolite accumulation and detoxification of ROS under water stress conditions. Overall, our results demonstrated that GB-priming improved the physiological and biochemical attributes of wheat plants under WS conditions by improving the drought perception capacity of wheat roots, ultimately enhancing the water stress tolerance. Thus, the GB-priming of roots could help to enhance the water-stress tolerance of economically important crops (i.e., wheat).

**Keywords:** osmoprotectants; osmotic stress; plant priming; root architecture; transcriptional factors; ABA-responsive genes

## 1. Introduction

Wheat (*Triticum aestivum* L.) is one of the most important staple crops that contributes significantly to world food security. However, water deficits induced by climate change pose a significant threat to wheat production in many regions of the world. Water deficits

substantially impact plant water status, root growth, hormonal homeostasis, net photosynthetic efficiency, lipid peroxidation, and the biological membrane, resulting in a significant reduction in biomass and yield [1,2]. Roots provide anchoring and facilitate the acquisition of water and minerals from the soil to support the aerial parts. Roots are the first organs that sense water deficit in soil and interact directly with edaphic water. They have an important role in plant adaptation to water deficits. Therefore, a well-developed root architecture system could determine plant stress tolerance and productivity under water deficits [3,4].

Priming is an effective strategy that stimulates plant stress response and memory to combat biotic and abiotic stresses [5]. Plants can respond to the environmental stimulus by modifying the phenotypic characteristics, metabolic, subcellular and proteomic activities, and transcriptional levels to reprogram the expression of microRNAs [6–9]. Various chemical compounds such as amino acids (proline, glycine betaine [10,11]), hormones (salicylic acid, abscisic acid [12,13]), minerals (selenium, calcium chloride [14]), and reactive oxygen species [15] have shown the ability to serve as a defense priming agent against environmental stresses [16]. Lower concentration of these priming agents increases seed germination, seedling growth, dry matter accumulation, better root architecture, and improved abiotic stress resilience [17–19]. In contrast, the higher concentration is often correlated with poor root growth, decreased carbon fixation, reduced antioxidant activity, increased lipid peroxidation, ion leakage, and photoinhibition, severely affecting crop productivity [20–25]. There has been an increased recognition that more attention needs to be paid to finding suitable priming agents with optimum concentrations, and adaptive strategies and modes of application for robust stress tolerance in crop plants.

Various techniques are applied to augment plant stress tolerance against different environmental stimuli. Among these, exogenous application of glycine betaine (GB) is a widely practiced technique. GB increases the plant stress tolerance via activation of the antioxidant system, protection of cellular structures, detoxification of ROS, and enhanced photosynthetic efficiency [10,26–28]. However, the gene regulatory mechanisms under the GB-induced stress tolerance are still poorly elucidated. Particularly, stress perception mechanisms leading to transcriptional changes and the signal transduction pathways that are involved in GB-induced plant stress tolerance still need to be studied in detail. Previous studies were conducted either on seed priming [28–30] or foliar application [11,26,31,32] on aerial parts of the plants. However, in the current research, we used GB-mediated root-priming technique to evaluate the physiological, morphological, hormonal and transcriptional changes in the roots of wheat seedlings. This is a novel area of research, and little is known regarding the GB-mediated root priming. This study provides insight into GB-mediated root morphological changes, ABA metabolism and its downstream signaling transduction, activation of transcriptional factors, and their roles in physiological regulation and water-stress tolerance in wheat.

## 2. Materials and Methods

This research was performed at the Plant Growth Regulator Engineering Research Centre, College of Agronomy & Biotechnology, China Agricultural University, Beijing, China, in 2018–2019. The wheat seeds (*Triticum aestivum* L. cv. Jimai-22, bred and provided by the Shandong Academy of Agricultural Science) of uniform size were surface-sterilized for 10 min with 3%NaClO. Seeds were placed on plastic germination plates containing distilled water (the lower surface of seeds was immersed in water). After completing germination (fourth day), the solution was changed with 1/4th concentration of Hoagland solution (HS) [31] and replaced with a new solution after every four days. At the three-leaf stage (about 12 days after germination), uniform seedlings were selected to execute the experimental treatments. Seedlings were removed from germination plates, rinsed thoroughly with distilled water, and treated with HS + 0 mM GB (GB-0) as control, and HS + 50 mM (GB-1) and HS + 100 mM (GB-2) as GB-primed treatments for 48 h. After GB-priming, drought stress was imposed by using 15% PEG-6000. Taken together, in this

split-plot layout, two conditions, namely well-watered (WW) and water-stress (WS), were maintained as main plots while three treatments (namely GB-0, GB-1, and GB-2, were maintained as sub-plots). Water stress (WS) was mainly imposed for six days, except for the genetic experiments in which water stress was imposed for 24 h (WS-1) and 48 h (WS-2). Plants were grown and kept under a greenhouse under controlled environment (22/18 °C day/night temperature, 12-h photoperiod, 60% relative humidity and 650 mol m$^2$ s$^1$ light intensity) during the whole study period. The GB (Cas: 107-43-7) were obtained from Sigma-Aldrich (Shanghai, China).

### 2.1. Plant Growth and Biomass Accumulation

After six days of treatments, seedlings were harvested to get fresh and dry weight of shoot and root. Samples were dried in oven (SANYO, Model no: MOV-202, Osaka, Japan) for 72 h at 80 °C.

### 2.2. Root Characteristics

Three individual plants were chosen from each treatment to record the root traits by placing each plant's roots in a 20 cm × 15 cm × 2 cm scanning tray. Roots overlapping was avoided by the precise arrangement of roots in the scanning tray. Roots were scanned using an Epson image scanner (Epson Perfection V700 with a precision of 6400 dpi) (Epson, Long Beach, CA, USA). The total root length, root volume, and root surface area of each treatment were calculated using WinRHIZO Pro image analysis software (Regent Instruments, Inc., Quebec City, QC, Canada).

### 2.3. Root Antioxidant Enzymes Extractions and Assays

The methods defined by Seckin et al. [32] were followed for antioxidant enzymes extraction with some minor modifications. Root specimens were collected from GB primed and non- primed plants grown in well-water and water-stressed conditions, washed thoroughly with distilled water to remove traces on roots, frozen in liquid nitrogen, and brought into the laboratory to store at −80 °C for subsequent analysis. 0.5 g root samples were taken and extracted into a prechilled mortar containing 50 mmol/L sodium phosphate (pH 7.8), 1 m EDTA, 50 mmol of magnesium chloride, 1 mM EDTA-Na2 and 1% (*w/v*) PVPP. The root extract was centrifuged at 12,000× *g* for 25 min, supernatant was collected and used to determine antioxidant enzyme quantifications.

Superoxide dismutase activity was determined by using Xue et al. [33] methods with slight modifications. A reaction mixture of 50 µL enzyme extract, 1.5 mL of 50 mM phosphate buffer (pH 7.8); 300 µL of 130 mM methionine, 300 µL of 0.750 µm Nitroblue Tetrazolium (NBT); 300 µL of 100 µM EDTA, 300 µL of 20 µM riboflavin and 250 µL of deionized water were prepared in a 10 mL transparent centrifuge tube. The test tube containing the reaction mixture was exposed to fluorescent light (40 mol m$^{-2}$ s$^{-1}$) for 20 min. The absorbance of the samples was assessed with a UNICO UV-2800AH spectrophotometer (UNICO Instrument Co. Ltd., Shanghai, China) at 560 nm wavelength.

Peroxidase (POD) activity was determined by the protocol proposed by Chance and Maehly [34] with minor modifications. POD reaction solution was prepared with 50 mM phosphate buffer with pH 7, 2.93 mL of 50 mM guaiacol, 50 µL of 0.05 percent $H_2O_2$, and 20 µL enzyme extract for measuring peroxidation of $H_2O_2$ by using guaiacol as an electron donor. After every second, the increase in absorbance due to the formation of tetraguaiacol was recorded at 470 nm. One unit of enzyme activity was calculated to be the rate of the enzyme responsible for a 0.01 increase in OD value in one minute. The enzyme activity was assayed and calculated as a unit mg$^{-1}$ protein basis.

### 2.4. Quantification of the Malondialdehyde (MDA) and Hydrogen Peroxide ($H_2O_2$) Content

The lipid peroxidation in roots was assessed by the thiobarbituric acid (TBA) test which indicates MDA as a final product of lipid peroxidation [35]. Approximately 500 mg of root tissue were homogenized in 5 mL of 0.1 percent (*w/v*) Trichloroacetic acid (TCA) solution.

After centrifuging the homogenate at $10,000 \times g$ for 20 min, 0.75 mL of the supernatant was transferred to 1.5 mL of 0.5 percent ($w/v$) TBA in 20 percent TCA. After 30 min of incubation in boiling water, the process was halted by immersing the reaction tubes in an ice bath. The samples were centrifuged at $10,000 \times g$ for 5 min, and the absorbance of the supernatant was recorded at 532 nm wavelength. At 600 nm, the value for non-specific absorption was subtracted. The extinction coefficient 155 mM$^{-1}$ cm$^{-1}$ was used to measure the amount of MDA–TBA complex (red pigment), MDA (nmol g$^{-1}$ fresh weight) = (Abs532 -Abs600)/155.

The method developed by Alexieva et al. [36] for measuring $H_2O_2$ was used in this study to quantify $H_2O_2$ content in root samples. 5 mL of 0.1 percent ($w/v$) Trichloroacetic acid (TCA) was used to homogenize 500 mg of root sample in pre-chilled mortar. After homogenization, the sample tubes were centrifuged at $10,000 \times g$ for 20 min. After centrifugation, 0.5 mL of the supernatant was combined with 0.5 mL of 10 mM potassium phosphate buffer (pH 7.0) and 1 mL of 1 Molar potassium iodide (KI) was used to quantify the absorbance of the supernatant at 390 nm wavelength, and $H_2O_2$ content was plotted on a standard curve.

### 2.5. Quantification of Proline and Total Soluble Sugars

Extraction of free proline content from the roots of wheat seedlings was carried out with 3% sulfosalicylic acid followed by centrifugation at $12,000 \times g$ for 10 min. Two ml of root extract combined with 2 mL of glacial acetic acid and 2 mL of acidic ninhydrin solution were mixed and incubated in the hot water bath at 100 °C for 30 min. After 30 min, the reaction was terminated by placing reaction tubes in an ice bath. The chromophore was isolated with five mL of toluene, warmed at room temperature, and the absorbance of chromophore was recorded at 520 nm wavelength [37].

Root samples that had been frozen at −80 °C were used to isolate total soluble sugars. Approximately 0.5 g of root sample were homogenized in 5 mL of distilled water and incubated in a hot-water bath at 100 °C for 20 min, the homogenate was centrifuged at $12,000 \times g$ for 10 min, and the supernatant was transferred to a 15 mL tube, the extraction repeated twice, and the supernatants were gathered. The supernatant was combined with Anthrone reagent, heated for 7 min, and the process was stopped by placing reaction tubes in an ice bath for 10 min. The absorbance of the solution was measured at 625 nm wavelength [38] and represented as mg g$^{-1}$ fresh weight.

### 2.6. Quantification of ABA, SA and JA and in Wheat Roots

After six days of 15% PEG stress, the root specimens were washed off with deionized water and transferred immediately to liquid nitrogen and saved for further hormonal quantification. Approximately 50 mg of root sample was powdered with liquid nitrogen, and then extraction proceeded with 500 μL of isopropanol for UPLC-TOF-MS analyses. The crude extracts were dissolved into 100 μL MeOH/$H_2O$ (85:15, $v/v$) and dried through nitrogen at 40 °C and filtrated by Sep-Pak Vac 3 mL C18 to remove color component interfering polar compounds. 100 μL MeOH/$H_2O$ (85:15, $v/v$) were used to isolate the compounds of interest in the cartridge. The concentrated sample was dissolved in 200 uL MeOH/$H_2O$ (85:15, $v/v$) for UPLC-TOF-MS (Agilent LC/MS QQQ, Palo Alto, CA, USA) analyses [39,40].

### 2.7. RNA Isolation, cDNA Synthesis, and Quantitative PCR Analysis

The total RNA was isolated from root tissues of GB-treated and non-treated plants subjected to well-watered and water-stress conditions by using TRIZOL® reagent (Invitrogen, Carlsbad, CA, USA) RNA extraction kit and purified by using Qiagen RNeasy columns (Qiagen, Hilden, Germany) according to the manufacturer's instruction. The quality of isolated RNA was assessed by using a Nanodrop spectrophotometer (Nanodrop Technologies, Rockland, DE, USA). The isolated RNA was primed with an oligo dT primer for cDNA synthesis using the Superscript III reverse transcriptase enzyme. Real-time

quantitative PCR was performed on a 7500 real-time PCR device (Applied Biosystems, Carlsbad, CA, USA) using SYBR® Premix Ex Taq™ (Takara, Japan). Three similar qPCR mixture system was used to construct three technical replicates for each biological replicate. The supplementary Table S1 presents the gene-specific primers generated with DNAMAN software (Lynnon, Quebec, QC, Canada). To normalize the data, the Actin-1 gene was selected as the internal reference gene. The cycle threshold (Ct) values were used to calculate the relative expression levels of all selected genes computed with the formula $2^{-\Delta\Delta Ct}$, which were then analyzed using one way ANOVA ($p < 0.05$).

### 2.8. Statistical Analysis

Microsoft Excel 2016 (Microsoft Corporation, Redmond, WA, USA) and Statistix 8.1 (Analytical Software, Tallahassee, FL, USA) softwares were used for data computations and statistical analysis. The least significant difference (LSD at 0.05) test was used to differentiate significantly different treatment means.

## 3. Results

### 3.1. Root Characteristics

Morphological characteristics of GB-primed (GB-1, GB-2) and non-primed (GB-0) wheat seedlings grown under well-watered (WW) and water-stressed (WS) conditions are presented in Figure 1a,b. Water stress significantly inhibited root length, root volume and surface area (Figure 1c–e). However, 48 h of GB-priming alleviated these deleterious effects of water stress. GB-mediated root priming increased total root length, root volume and surface area as compared to non-primed plants in both WW and WS conditions. Under WW conditions, GB-2 showed 35.4, 24.8, and 42.7% greater root length, surface area and root volume than GB-0, respectively. Consistently, under WS conditions, GB-primed plants showed lower susceptibility to water stress. For instance, in contrast to 42.0, 45.2, and 54.0% reduction of total root length, surface area and root volume in GB-0 plants, GB-2 plants showed only 8.3, 12.5, and 13.5% reduction as comparesd to their relevant control, respectively. These results suggest the higher susceptibility of GB-0 plants to osmotic stress. Moreover, water stress not only affected the root characteristics but also reduced shoot length (16.4%) in GB-0 treatment. Whereas GB-2 primed plants showed better performance, with 16.4 and 2.3% higher shoot length than the GB-0 non-primed plants under well-water and water stress conditions, respectively (Figure 1f).

### 3.2. Wheat Seedling Fresh Weight and Dry Matter Accumulation

Water stress significantly reduced the growth attributes of wheat seedling, but GB-priming protected the wheat seedlings against the damaging effects of water stress. Under WW conditions, the maximum values for fresh and dry weights of root and shoot were observed. Notably, the GB-primed treatment GB-2 always showed the highest values followed by GB-1 and GB-0 (Figure 2a–d). Consistently, under WS conditions, GB-primed plants (GB-1, GB-2) showed a significantly lower reduction in shoot and root weights (both fresh and dry) than the non-primed (GB-0) plants. The least reduction in shoot and root dry matter accumulation was recorded in GB-2 (4.4% and 7.4%), followed by GB-1 (8.8 and 17.3%) and Gb-0 (32 and 30%), suggesting that GB-mediating priming could maintain the growth of wheat seedlings under water stress conditions.

#### 3.2.1. Antioxidant Enzymes Activity

The activities of antioxidant enzymes named superoxide dismutase (SOD) and peroxidase (POD) in the roots of wheat seedlings were increased under water stress conditions as compared to well-water conditions (Figure 3a,b). GB-mediated root priming with 100 mM concentration (GB-2) significantly upregulated the activities of SOD and POD (416 and 142%) in the roots as compared with the non-primed (197 and 90%) plants grown under water stress conditions. Eventually, it augmented substantial protection against oxidative damage induced by ROS overaccumulation.

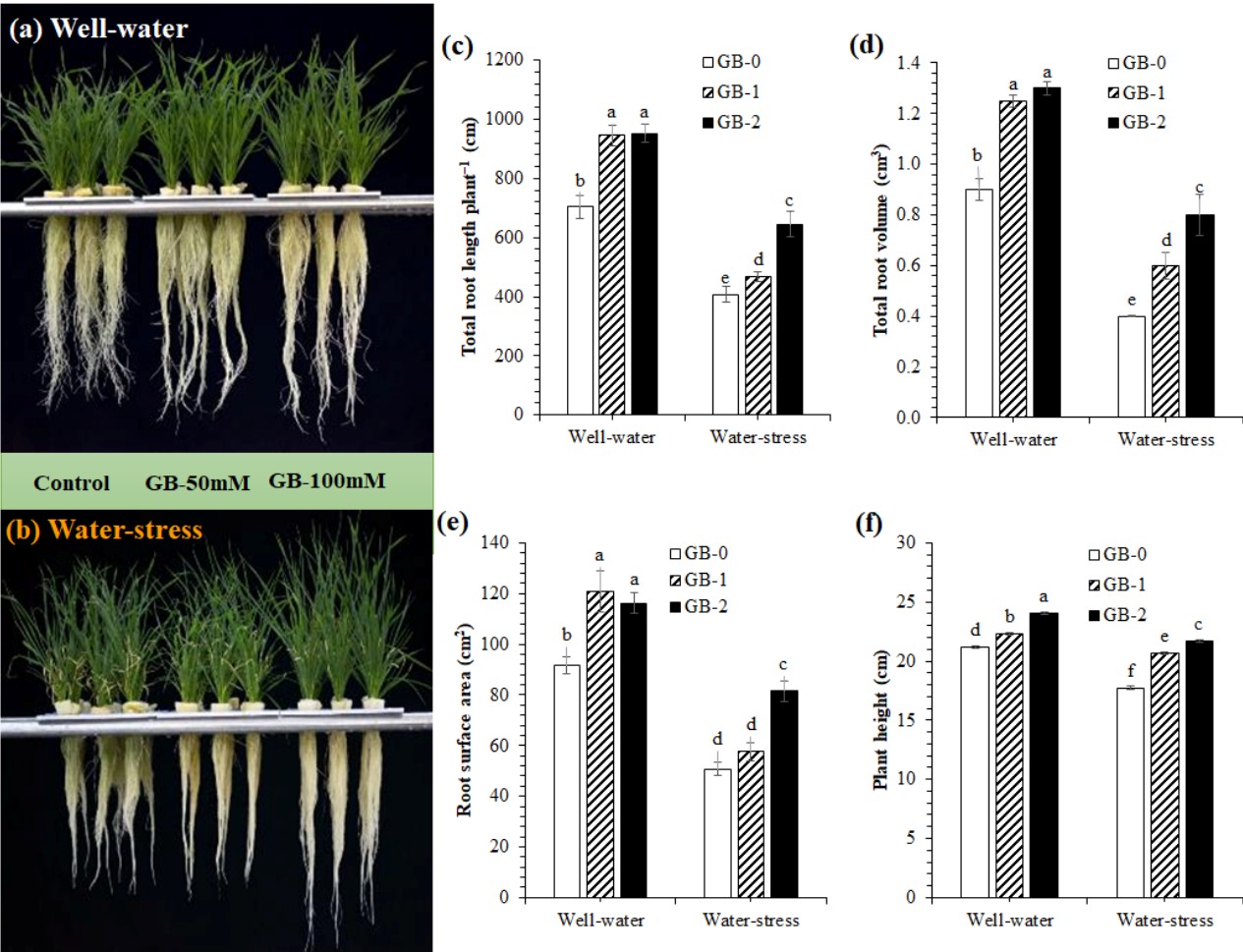

**Figure 1.** Effect of GB root-priming on wheat seedling morphological characteristics (**a**,**b**), root architecture (total root length (**c**), root volume (**d**), surface area (**e**) and shoot length (**f**) of wheat seedling grown under well-water and water-stress conditions. Each value represents the mean ± standard deviation (SD) of three replicates. Bars showing different letters represent the significant differences at $p \leq 0.05$ as determined by LSD test.

### 3.2.2. MDA and $H_2O_2$ Accumulation

The lipid peroxidation measured as MDA content and generation of ROS assessed as $H_2O_2$ content in the roots of primed and non-primed wheat seedlings grown under well-watered and water-stressed conditions are shown in Figure 3. No significant difference was observed between GB-primed and non-primed plants under normal or well-watered conditions regarding the MDA and $H_2O_2$ levels. However, PEG-stress (WS conditions) drastically triggered the MDA and $H_2O_2$ accumulation in the roots of wheat seedlings, more obviously (50 and 58%) in GB-0 plants. Importantly, the GB-mediated root priming effectively alleviated the PEG-induced oxidative damage as GB-2 roots accumulated 33 and 11% higher amounts of MDA and $H_2O_2$ than GB-0 of WW, suggesting that priming of wheat roots with GB could protect the wheat plants by reducing the overaccumulation of ROSs under water stress conditions.

### 3.3. Osmolytes Accumulation

Under well-watered conditions, the content of osmolytes such as proline and total soluble sugars did not show any significant difference in the roots of GB-primed and non-primed plants (Figure 4a,b). However, water stress induced the accumulation of proline and soluble sugars in the root of GB-0, GB-1, and GB-2 plants. Compared to GB-0 plants of WW, GB-0 plants accumulated significantly higher (612 and 249%) amounts of

both osmolytes than GB-2 (412 and 196%) plants grown under WS conditions, suggesting that GB-mediated priming might alter the osmotic potential of wheat roots, affecting the biosynthesis and accumulation of osmolytes.

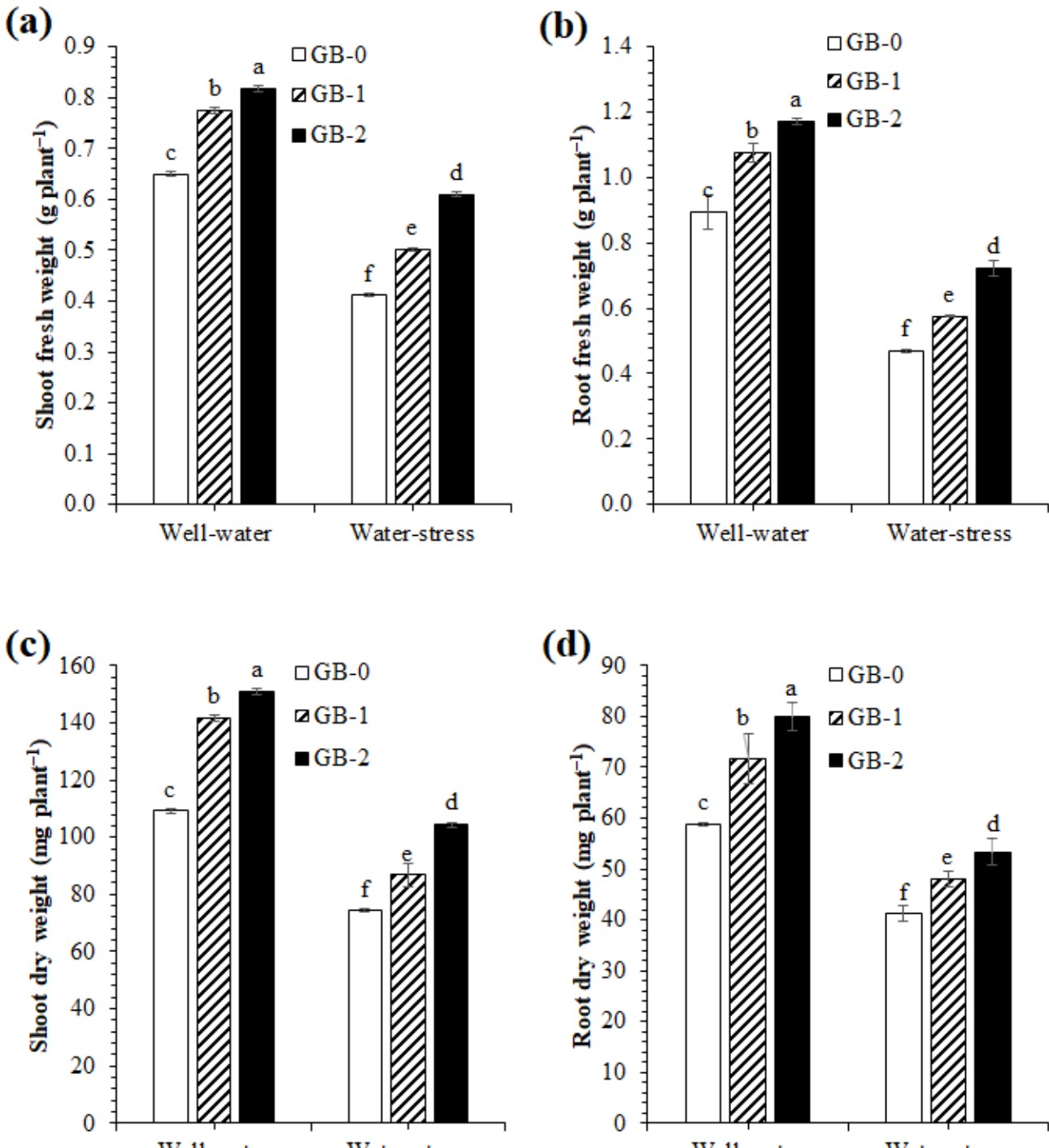

**Figure 2.** Effect of GB root-priming on shoot (**a,c**)/root (**b,d**) fresh weight and dry matter accumulation of wheat seedling grown under well-water and water-stress conditions. Each value represents the mean ± standard deviation (SD) of three replicates. Bars showing different letters represent the significant differences at $p \leq 0.05$ as determined by LSD test.

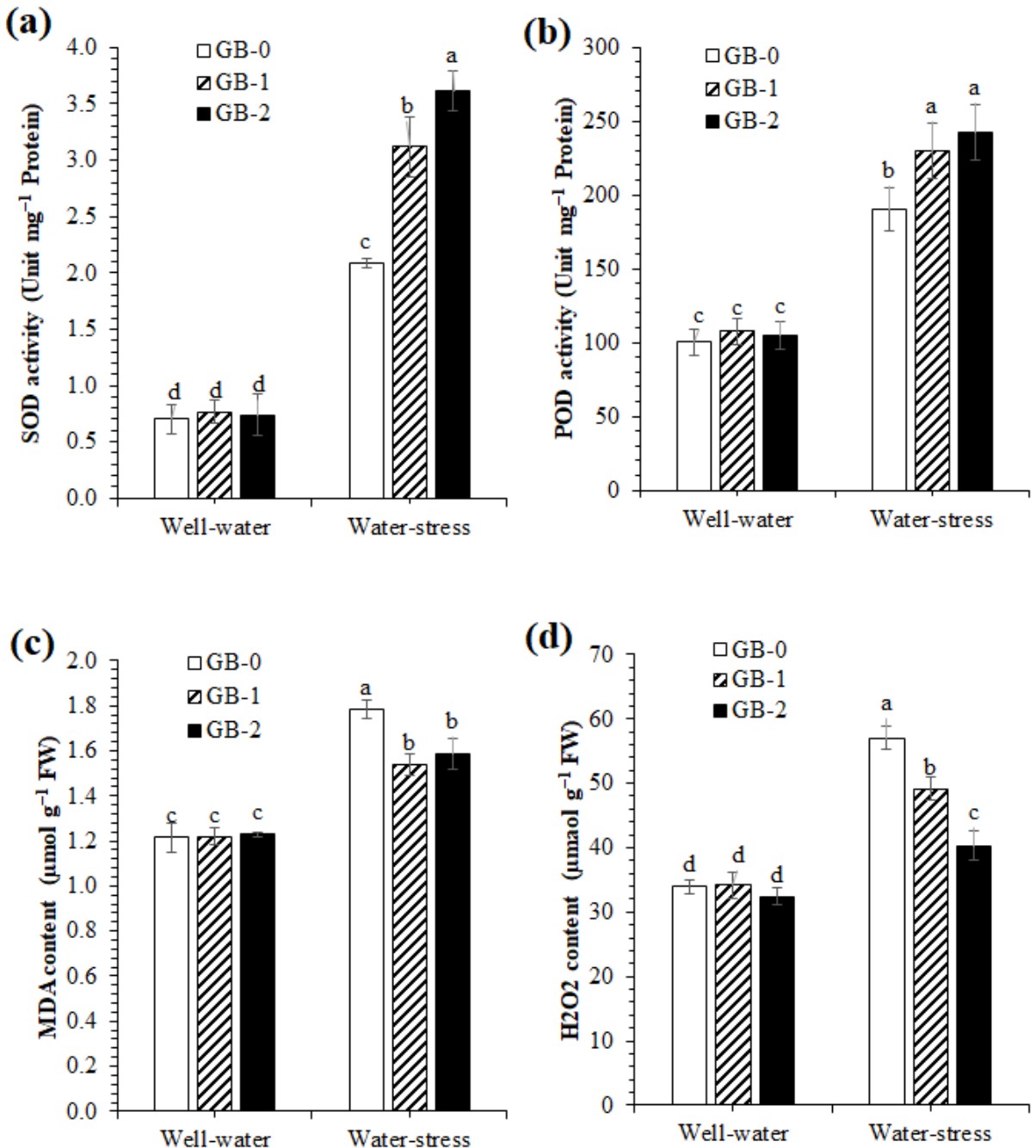

**Figure 3.** Effect of GB root-priming on antioxidant (SOD, POD) activities (**a**,**b**) and MDA, $H_2O_2$, content (**c**,**d**) in the roots of wheat seedling grown under well-water and water-stress conditions. Each value represents the mean ± standard deviation (SD) of three replicates. Bars showing different letters represent the significant differences at $p \leq 0.05$ as determined by LSD test.

### 3.4. Hormones

The pattern of endogenous accumulation of stress-responsive hormones (ABA, SA, and JA) in the roots responded differently in GB-primed and non-primed wheat seedling grown under well-watered and water-stressed conditions, as shown in Figure 5. There was no significant difference in levels of hormones between GB-0, GB-1 and GB-2 under WW, except SA, which showed significantly higher (38%) accumulation in GB-2 plants. Initially, the levels of four hormones were significantly lower under WW. However, WS conditions triggered the accumulation of ABA, SA, and JA in the roots of wheat seedlings. Importantly,



plants primed with 100 mM of GB (GB-2) accumulated a significantly higher amount of ABA and SA (122 and 86%) than non-primed GB-0 plants. In contrast, we did not find any significant variation in JA accumulation in the roots of primed and non-primed plants.

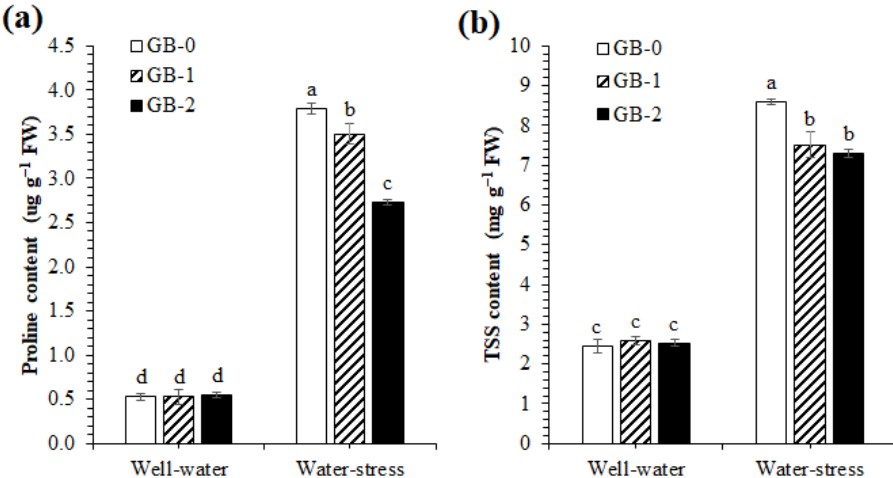

**Figure 4.** Effect of GB root-priming on proline (**a**) and total soluble sugars content (**b**) in the roots of wheat seedling grown under well-water and water-stress conditions. Each value represents the mean ± standard deviation (SD) of three replicates. Bars showing different letters represent the significant differences at $p \leq 0.05$ as determined by LSD test.

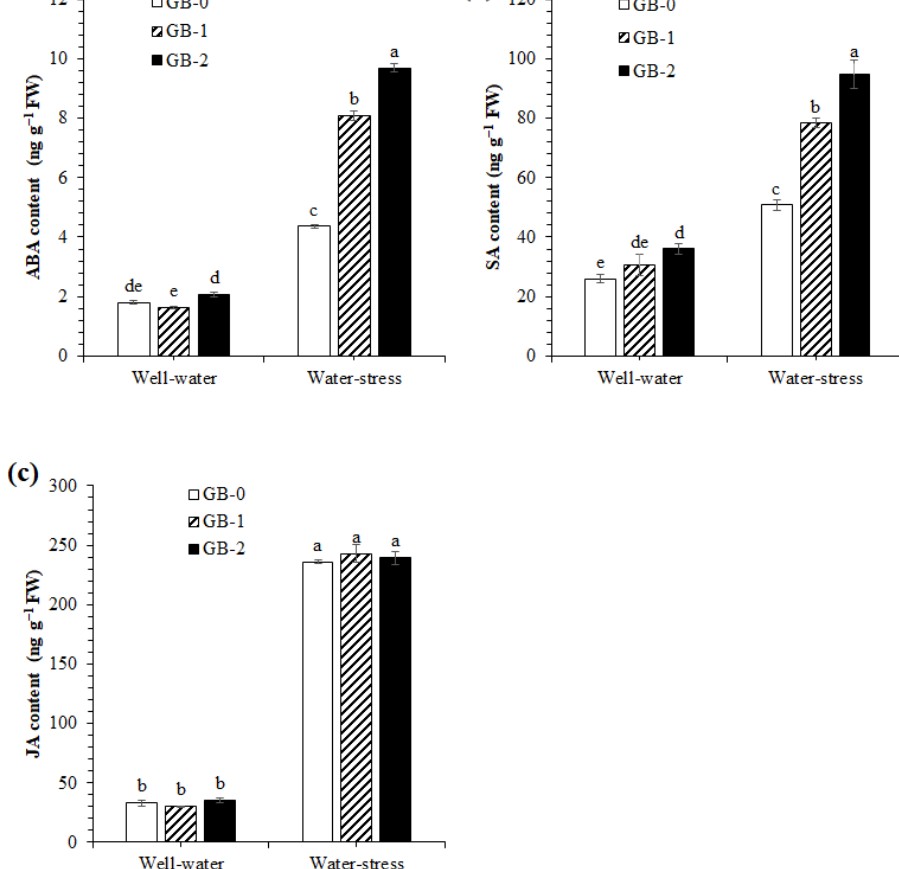

**Figure 5.** Effect of GB root-priming on hormones accumulation (ABA, SA, and JA) in the roots of wheat seedling grown under well-water and water-stress conditions (**a–c**). Each value represents the mean ± standard deviation (SD) of three replicates. Bars showing different letters represent the significant differences at $p \leq 0.05$ as determined by LSD test.

*3.5. Changes in Expression of Genes Involved in ABA Metabolism*

　　　　To better understand the modulation of endogenous accumulation of ABA between GB-primed (GB-2) and non-primed (GB-0) roots of wheat seedlings, we investigated the expression pattern of genes involved in ABA biosynthesis (*TaNECD-1, 9-cis-epoxycarotenoid dioxygenase*) and degradation (*TaABA8′OH2, 8′-hydroxylase*) under 24 h (WS-1) and 48 h (WS-2) of PEG-induced water stress conditions. Osmotic stress-induced the linear increase in the expression of both the genes (*TaNECD-1; TaABA8′OH2)* involved in ABA metabolism in all treatments (Figure 6a,b). However, the GB-2 plants always showed significantly higher expression levels than GB-0, while the highest expression of both the genes was recorded under WS-2, suggesting the direct or indirect association between GB and ABA metabolism.

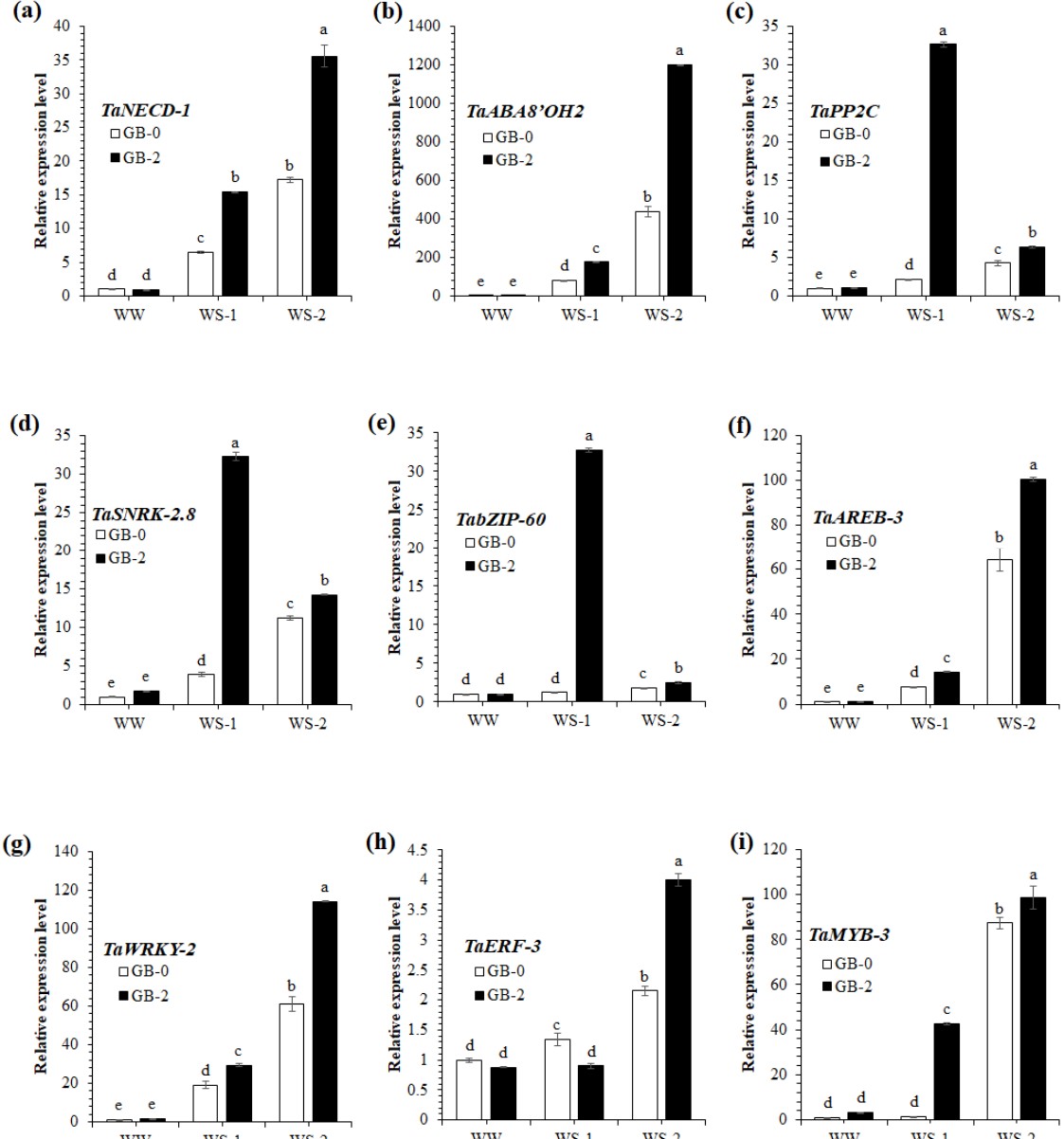

**Figure 6.** Effect of GB root-priming on expression levels of *TaNECD-1* (**a**), *TaABA8′OH2* (**b**), *TaPP2C* (**c**), *TaSNRK-2.8* (**d**), *TabZIP-60* (**e**), *TaAREB-3* (**f**), *TaWRKY-2* (**g**), *TaERF-3* (**h**), and *TaMYB-3* (**i**) in the roots of wheat seedlings in well-water (WW), after 24 h water-stress (WS-1) and after 48 h water-stress (WS-2). Each value represents the mean ± standard deviation (SD) of three replicates. Bars showing different letters represent the significant differences at $p \leq 0.05$ as determined by LSD test.

### 3.6. Changes in Expression of Genes Involved in ABA Downstream Signaling

Water-stress mediated ABA signaling initiated by ABA interaction with *PP2C;* this interaction plays an important role in the phosphorylation of *SnRK2* for downstream regulation of ABA-dependent signaling cascade to activate transcriptional factors. Consistently, osmotic stress induces the expression of *TaPP2C* and *TaSNRK2.8* in the roots of both primed and non-primed plants (Figure 6c,d). However, the expression pattern of both *TaPP2C* and *TaSNRK2.8* was different between GB-0 and GB-2. Interestingly, non-primed plants showed increased expression of *TaPP2C* and *TaSNRK2.8* with the increasing exposure of osmotic stress. In contrast, GB-primed plants showed a sharp initial increase under WS-1 (24 h of PEG) followed by a significant decline under WS-2. Notably, despite the declining trend under WS-2, GB-primed roots always showed the highest expression levels of *TaPP2C* and *TaSNRK2.8* under water stress conditions.

### 3.7. Changes in Expression Patterns of Different Transcriptional Factors

Transcriptional factors (TFs) are important regulators of downstream gene regulation and abiotic and biotic stress resistance. GB mediated root priming significantly increased the expression level of studied TFs (*TabZIP-60, TaAREB-3, TaWRKY-2, TaERF-3,* and *TaMYB-3)* as compared to non-primed plants under water stress. The expression level of *TabZIP-60* in GB-primed roots inclined significantly under WS-1 and then declined under WS-2 (Figure 6e), while in the roots of non-primed seedling, it inclined only under WS-2 but remained significantly lower than GB-primed roots. The expression level of *TaAREB-3, TaWRKY-2, TaERF-3,* and *TaMYB-3* showed an increasing trend with time in both primed and non-primed roots. Notably, compared to non-primed plants, GB-mediated root priming significantly increased the expression level of *TaAREB-3, TaWRKY-2, TaERF-3,* and *TaMYB-3* (Figure 6f–i) under water stress conditions.

## 4. Discussion

Drought, as one of the main challenges that climate change has imposed on crop production, causes over 30% yield losses in agriculture [41]. Wheat is also very exposed to such yield losses as it is an important staple food worldwide and widely grown in drought-prone environments [42]. Therefore, to shield the wheat against such yield losses and maintain its productivity for food security, it is very crucial to improve the drought tolerance of wheat [43]. From this perspective, the improvement in the response of wheat roots under water shortage could be a useful strategy to protect plants against drought stress, as roots are indeed the water pumps for plants. One of the modern agricultural strategies for improvement of stress response in plants include the application of exogenous materials particularly the solutes that are already compatible with plants, such as phytohormones, proline and glycine-betaine (GB). Keeping in view the significant consideration of exogenous materials in improving the stress responsiveness of plants, we also examined the effects of GB-mediated root priming of wheat plants under well-watered (WW) and PEG-mediated water-stressed (WS) conditions. Studies reporting this particular mechanism are minimal. Notably, our results revealed that priming of wheat roots with glycine-betaine (GB) could significantly improve the growth and overall response of wheat roots, especially under water-stress (WS) conditions. Consequently, roots effectively countered the drought-induced oxidative stress as suggested by declined accumulation of reactive oxygen species (ROS) and upregulated activity of major antioxidant enzymes (SOD, POD), drought-responsive hormones (ABA, SA, JA) and the underlying molecular drivers (i.e., genes and transcription factors). Importantly, our results provide an important as well as novel insight into the mechanism of drought response of wheat roots by the GB priming.

The morphological attributes of roots are very crucial to avoid dehydration in the plants through efficient uptake of water and nutrients under drought stress [44]. It ensures the survival of plants by maintaining the osmotic potential, vegetative growth and biomass accumulation [44]. Thus, it is necessary to promote an efficient root system capable of

exploiting residual soil moisture under suboptimal conditions and mitigating oxidative stress caused by low to moderate water stress [45]. In the present study, drought significantly reduced the growth attributes of roots including total length, volume, and surface area, followed by the impaired shoot growth. However, the GB-primed plants (GB-1, GB-2) showed significantly higher values of root and shoot parameters than non-primed plants (GB-0) under both WW and WS conditions. The significantly reduced growth and surface area of roots under WS conditions might be due to the severe decline in turgidity and cell expansion at root meristem that caused the shrinkage of root length, surface area, and root proliferation [46]. As a result, it hampered the absorption of important minerals required for proper plant growth, causing stunted growth and wilting of wheat seedlings and reduced biomass [45]. Previously, a 4% decline in root biomass of wheat resulted in a 20% decline in shoot biomass under drought, indicating the severe consequences of impaired root growth on the vegetative growth of plants under drought stress [47].It suggests that the comparatively better growth of GB-primed plants, particularly the GB-2, could be attributed to their better root growth under WS conditions that ultimately resulted in a better supply of nutrients to vegetative parts and maintained the integrity of wheat seedlings. Consistent with the trends in our study, it has been shown that exogenous application of GB enhances dry matter accumulation and stress tolerance under abiotically stressed environment [11,30,31]. Our results suggest that priming of wheat roots with 100 mM of GB significantly ameliorates the impact of drought on the wheat seedlings by maintaining the root growth under water-limited conditions.

To further investigate the reasons for the growth difference between the GB-primed and GB-0 plants, we estimated the extent of oxidative stress by measuring MDA content in the roots of wheat plants. Under abiotic stresses, uncontrolled free radicals are significantly increased and the concentration of antioxidants, nonenzymatic and enzymatic, is enhanced to regulate cellular homeostasis and reduce lipid peroxidation of the plasma membrane [48]. Reactive oxygen species (ROS) such as $H_2O_2$ are important signaling molecules, but their over-production in plants under stressful environments might lead to serious oxidative damage, causing damage to cellular membranes by increasing lipid peroxidation [26,49] Therefore, the accumulation of ROS and MDA contents in cells could be utilized as common markers for oxidative damage in plants. Consistent with the morphological attributes, the results of MDA and $H_2O_2$ quantification revealed that non-primed roots (GB-0) accumulated significantly higher amounts of these compounds as compared to GB-primed (GB-2) roots, suggesting that priming of roots with GB reduced the drought-induced oxidative damage on wheat plants that resulted in a better response of GB-primed plants under the WS conditions. It was further confirmed by the significantly higher activities of SOD and POD in the GB-primed roots, which suggests that priming of roots with GB enables the wheat plant to detoxify the drought-induced ROS accumulation by promoting the activities of major antioxidant enzymes. Similarly, in a previous study, the drought-tolerant wheat presented the higher activities of antioxidant enzymes, and less ROS accumulation and oxidative damage than drought susceptible wheat [49]. Therefore, the lower oxidative damage and better antioxidative capacity of GB-primed roots suggest that GB-mediated priming of wheat roots enhances the tolerance of wheat against drought stress. Our results are consistent with the previous studies with exogenous application of GB [50–52].

Plants carry out osmotic adjustments under drought stress by promoting the accumulation of severe low-molecular osmolytes such as proline and soluble sugars [53]. Similarly, the WS conditions induced a significant accumulation of proline and soluble sugars in the roots, particularly in non-primed plants (GB-0). Previously, the sensitive genotypes of wheat accumulated a higher concentration of soluble sugars and proline than tolerant genotypes under water stress conditions [54]. Accordingly, in the present study, the lower levels of osmolytes in GB-2 suggest the better osmotic potential of GB primed roots under drought stress and are consistent with the less oxidative damage and better growth of GB primed plants than non-primed plants. In addition, GB is also considered an important osmoprotectant molecule that could maintain osmotic balance under stressful environ-

ments [55]. Therefore, the lower levels of proline and soluble sugars and better osmotic potential in GB-primed plants could also be attributed to better GB availability. Altogether, it suggests that GB priming of wheat roots reduces their susceptibility towards drought stress and alters the drought signaling for osmolyte accumulation.

Phytohormones are critical for regulating various responses of plants that help to acclimatize the changing environmental conditions [55]. Previously, studies had reported the accumulation of these hormones such as abscisic acid (ABA), salicylic acid (SA) [56], auxins (Indole acetic acid; IAA) [57], Jasmonic acid (JA) [58], and ethylene (ETH) [59] that alleviated the abiotic stresses in plants. Similarly, in our study, WS conditions induced significant upregulation of all the studied hormones endorsing the critical role of ABA, SA and JA under water limiting conditions. Among all the phytohormones, ABA is considered the primary hormone that induces drought tolerance in plants by modulating the stomatal behavior, root growth and ABA-dependent signaling [55]. In addition to abscisic acid, JA, ETH, and SA are also involved in osmotic adjustments and other drought-related signaling to help plants survive the prevailing drought conditions [56,58]. Under normal conditions (WW), except for the significantly higher accumulation of SA in GB-2, GB-priming of roots did not induce any significant accumulation of phytohormones as compared to non-primed roots. It suggests that SA is more responsive towards the GB application. In this perspective, a recent study also reported that the combination of SA and GB is more robust in protecting the plants against water stress [60]. Therefore, it could be assumed that in present study, some sort of mutual synergy between SA and GB application helped the GB-primed plants to respond better under drought. Additionally, it has been well reported that SA protects plants against drought by inducing the ABA accumulation [61]. Therefore, the remarkably better growth of GB-primed plants, particularly under WS conditions, could be attributed to their higher amounts of ABA and SA than non-primed root. Taken together, these results suggest that GB-priming assisted the early perception of drought by triggering the higher accumulation of drought-responsive phytohormones that might modulate the stress signaling in GB-primed plants to maintain their growth under WS conditions.

To better understand the molecular mechanisms of GB-induced water stress tolerance in wheat seedlings, we investigated the effect of GB priming on the transcription level of genes involved in ABA metabolism, its downstream signaling and in the activation of transcriptional factors in the roots. *TaNCED* is considered the most prominent gene family involved in ABA biosynthesis [62], while *ABA8′OH2*, an ABA catabolic gene, predominantly regulates the endogenous levels of ABA in plants [63]. Previously, the expression levels of these genes have been linked to the drought adaptation of plants [63]. For example, the overexpression of *OsNCED3* and *OsABA8′OX* in rice induced the drought resistance by regulating ABA levels [64]. Consistently, the improved drought tolerance of GB-primed roots in our study could also be attributed to markedly higher expression of these two genes as compared to non-primed plants under WS conditions (WS-1, WS-2). It suggests that GB-priming modulated the ABA biosynthesis and its endogenous levels in the roots that induced the drought tolerance in wheat seedlings. Furthermore, ABA signaling under abiotic stress is recognized and transmitted to various downstream components. Among these components, protein kinases and phosphatases are considered as key components. Within protein phosphatases (PP), *PP2Cs* are essential components of ABA signaling to enhance stress resistance in plants [65]. It has been reported earlier that wheat *TaPP2C1* and rice *OsPP18* genes augmented the elevated drought and salinity tolerance due to ROS detoxification, better antioxidant capacity and activation of genes involved in the regulation of ABA-independent pathway [66]. Our results also showed that GB-priming triggered the *PP2C* expression in roots at 24 h of PEG stress that gradually declined at 48 h (Figure 6c). This initial overexpression of *TaPP2C* under WS-1 could be involved in regulating ROS homeostasis and activation of the plant antioxidant system, suggesting that GB-priming assisted the wheat plants in maintaining the redox balance by regulating both the ABA-dependent and ABA-independent pathways of drought tolerance. Besides phosphatases, protein kinases such as calcium-dependent protein kinases (CDPKs)

and SNF1-related kinases (SnRK) are also important downstream components of the ABA signaling network under abiotic stresses. Mainly, SnRK2 kinases are considered the main downstream regulator of ABA-signaling pathways and activated under drought and other abiotic stresses [67,68]. Previous reports articulated that upregulation of ABA-dependent and independent stress-responsive genes were attributed to overexpression of *TaSnRK2.3* and *TaSnRK2.8* in *Arabidopsis*, which enhanced resistance to drought, salinity, and cold stress [69]. Additionally, SnRK2 protein kinases can also influence the expression of osmotic stress-responsive genes by activating transcription factors [70]. Consistently, our results revealed that PEG stress induced the expression of *TaSnRK2.8*, and plants primed with GB showed higher *SnRK2.8* expression levels than their counterparts, suggesting that GB-primed plants had a higher capacity to regulate the stress-responsive genes or transcription factors to counter the drought-induced oxidative damage.

Transcription factors (TFs) play a crucial role in gene regulation and various TF families regulating abiotic stress responses are based in an ABA-dependent manner. These TFs mainly include bZIP, MYB/MYC, DBF (dehydration-responsive binding factor), NAC and WRKY family [71]. Keeping in view the expression pattern of ABA-signaling and biosynthesis genes, we predicted that GB-priming might also activate different TFs involved in osmotic stress tolerance. Therefore, we investigated the expression level of five members of different TFs families (*TabZIP-60, TaAREB-3; TaWRKY-2; TaERF-3; and TaMYB-3*) in the roots of wheat seedlings exposed to water-stress conditions (WS-1, WS-2). Overall, the GB-primed plants exhibited significantly higher expression levels of all the five transcription factors than non-primed plants, particularly under WS-2. This increased expression of TFs in GB-primed plants could be attributed to higher ABA accumulation and better capacity to perceive drought stress, ultimately resulting in a better response of GB-primed wheat seedlings under drought stress. Our findings are consistent with earlier studies reporting that the upregulation of *TaWRKY-2, TaERF3*, and *TaMYB-3* could improve abiotic stress resistance by enhancing plant antioxidant system, regulating ROS homeostasis and activating the multiple stress-responsive genes in downstream [71,72].

## 5. Conclusions

In the present study, water stress significantly impaired the growth and development of wheat seedlings. However, defense priming with GB improved the redox homeostasis of wheat roots by improving the antioxidative response and resulted in better detoxification of ROS and reduced oxidative damage, maintaining the better growth of wheat seedlings under drought stress. Moreover, GB-mediated root priming of wheat roots not only accelerated the biosynthesis and accumulation of stress-responsive phytohormones such as ABA and SA but also enhanced their downstream signaling that improved the root architecture systems and drought-sensitivity of wheat roots. Subsequently, it improved the drought tolerance of wheat seedlings by activating both ABA-dependent and ABA-independent pathways by upregulating the proteins kinases gene *TaSnRK2.8* and phosphatases gene *TaPP2C*. Ultimately, it regulated the downstream transduction of ABA-mediated stress signaling to activate different TFs genes (*TabZIP-60, TaAREB-3; TaWRKY-2; TaERF-3; and TaMYB-3*) that are directly or indirectly involved in the activation of plant antioxidant system, ROS scavenging and other developmental processes. Altogether, the series of experiments in our study unanimously concluded that priming of wheat roots with 100 mM of GB enhanced the sensitivity and responsiveness of wheat roots towards drought by improving the drought-induced signaling that allowed the wheat seedlings to effectively alleviate the harmful effects of drought. Therefore, our findings could provide the basis for further enhancing the growth and development of plants under osmotic stress conditions.

**Supplementary Materials:** The following are available online at https://www.mdpi.com/article/10.3390/agriculture11111127/s1. Supplementary information accompanies this paper are provided in supplementary Table S1.

**Author Contributions:** N.A and Y.Z conceptualize the research; N.A., M.Z. and Q.L. evaluated the physiological and biochemical analysis in roots of wheat seedlings; N.A. and Y.Z. performed the partial hormone assay, N.A. analyzed the data; Y.Z. provided some important suggestions for writing—original draft preparation; and N. A wrote the paper. X.W. and J.W. helped to revise the manuscript. All authors have read and agreed to the published version of the manuscript.

**Funding:** This research was supported by National Key Research and Development Program of China (Grant No. 2017YFD0300410).

**Institutional Review Board Statement:** Not applicable.

**Informed Consent Statement:** Not applicable.

**Acknowledgments:** We would like to express our special thanks to Muhammad Ali Raza and Atta Mohi Ud Din. They provided technical assistance for linguistic proofreading and data interpretation of our manuscript to finalize this manuscript.

**Conflicts of Interest:** The authors declare no conflict of interest.

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
