# Peer review of "Glycine Betaine-Mediated Root Priming Improves Water Stress Tolerance in Wheat (Triticum aestivum L.)"

_agriculture, doi:10.3390/agriculture11111127_

Round 1

Reviewer 1 Report

Water stress is a major global conundrum that severely impacts agricultural productivity and sustainability, particularly in arid and semiarid regions. The manuscript investigated a novel strategy using glycine betaine to alleviate the adverse impacts of water stress on soil. It is an interesting paper and the subject certainly falls within the general scope of Agriculture journal.

However, it is only recommended to revise the letters in Fig 1 and 2, its inappropriate and not reliable.

In addition, please add the physical and chemical properties before planting and after harvesting. 

I would like to add the yield related traits and productivity because its the end goal and to connect between root growth and vegetative growth response to GB application.

How to select the amounts of water for each growth stage?

Some writing errors should be checked, for example, the superscript and subscript Like H2O2 and etc....

In the section of Materials and Methods, some key information was missing in the description of experiment. Please check the whole section carefully and add those details. For example: How about the fertilization management?

Could you please let me know the source of glycine betaine and add its properties?

The writing could be improved by strengthening the connectivity between paragraphs. Factors tested in the study (i.e. water stress, GB, wheat, etc) are individually introduced and described in single paragraphs. There are several places where new topics are introduced and connections to the previous subject are not clear.

It is very important to apply this experiment in your field area to confirm these data,to keep the scope of the study as broad as possible

I prefer to mix between figures and tables in the results, not like your presentation

Author Response

We appreciate the reviewers’ comments and valuable suggestions. Followed by three Reviewers’ comments. We have revised the manuscript under the “track changes” mode. Our point-to-point responses are presented below.

Point1:It is only recommended to revise the letters in Fig 1 and 2, its inappropriate and not reliable.

Response: Thanks for this comment. We have revised the lettering in figures 1 and 2.

Point2:In addition, please add the physical and chemical properties before planting and after harvesting. 

Response: We used hydroponic culture for experiment therefore there is no need of physical or chemical properties what is required in in-vivo or soil culturing. Correspondingly, the Hoagland solution was used for hydroponic culture and the conponent was according to Hoagland and Arnon (1950) as mentioned in Line 90-91.

Hoagland, D.R.; Arnon, D.I. The water-culture method for growing plants without soil. Circ. Calif. Agric. Exp. Stn. 1950, 347, 1–32.

Point3:I would like to add the yield related traits and productivity because its the end goal and to connect between root growth and vegetative growth response to GB application.

Response: We sincerely thank the reviewer for valuable suggestions, in the current research developed novel root priming technique to evaluate effect of GB for water stress tolerance, In this research, we focused on the effects of GB on wheat water stress tolerance at seedling stage. However, we used glycine betaine for seed treatment  [1] and for exogenous application on wheat [2] and found positive contribution in crop yield and related attributes and our research has been published and reference is given below,

  1. Ahmed, N.; Zhang, Y.; Li, K.; Zhou, Y.; Zhang, M.; Li, Z. Exogenous application of glycine betaine improved water use efficiency in winter wheat (Triticum aestivum L.) via modulating photosynthetic efficiency and antioxidative capacity under conventional and limited irrigation conditions. Crop J. 2019, doi:10.1016/j.cj.2019.03.004.
  2. Ahmed, N.; Zhang, Y.; Hai, Y.; Gabar, A.; Zhou, Y.; Li, Z.; Zhang, M. Seed priming with glycine betaine improve seed germination characteristics and antioxidant capacity of wheat ( Triticum aestivum l .) seedlings under water-stress conditions. Appl. Ecol. Environ. Res. 2019, 17, 8333–8350.

Point4:How to select the amounts of water for each growth stage?

Response: As it was hydroponic culture and experiment duration was 15-18 days, for artificial stress condition we used PEG at 10% to create osmotic stress condition.

Point5:Some writing errors should be checked, for example, the superscript and subscript Like H2O2 etc....

Response: Thanks. We have revised as you suggested. At Line 147 and 260 the H2O2 replaced with H2O2. Similarly, in Line 184 and 188, H2O were replaced with H2O. Line 205 the formula ‘2−ΔΔCt’ were revised to ‘2−ΔΔCt’.

Point6:In the section of Materials and Methods, some key information was missing in the description of experiment. Please check the whole section carefully and add those details. For example: How about the fertilization management?

Response: Thanks. The fertilization of hydroponic culture media was managed according to Hoagland and reference is provided in Line 90-100

Point7:Could you please let me know the source of glycine betaine and add its properties?

Response: Sure. The Glycine betaine molecular weight 117.15, CAS Number: 107-43-7 were obtained from Sigma-Aldrich, Shanghai, China. These details were added in Line 108-109

Point8:The writing could be improved by strengthening the connectivity between paragraphs. Factors tested in the study (i.e. water stress, GB, wheat, etc) are individually introduced and described in single paragraphs. There are several places where new topics are introduced and connections to the previous subject are not clear.

Response: Thanks for the careful review of our manuscript. You mentioned a very critical point. According to your suggestion, we have carefully rechecked the punctuation and the meaningfulness of the sentences and improved it accordingly.

Point9:It is very important to apply this experiment in your field area to confirm these data, to keep the scope of the study as broad as possible.

Response: Thanks. We appreciated this Reviewer for proposing such great suggestions: In our previous studies we applied Glycine betaine in field condition under well-watered and limited irrigation conditions. It showed improved water use efficiency in winter wheat (Triticum aestivum L.) via modulating photosynthetic efficiency and antioxidative capacity under conventional and limited irrigation conditions.

Ahmed, N.; Zhang, Y.; Li, K.; Zhou, Y.; Zhang, M.; Li, Z. Exogenous application of glycine betaine improved water use efficiency in winter wheat (Triticum aestivum L.) via modulating photosynthetic efficiency and antioxidative capacity under conventional and limited irrigation conditions. Crop J. 2019, doi:10.1016/j.cj.2019.03.004.

Point10: I prefer to mix between figures and tables in the results, not like your presentation

Response: Thanks. Tables are generally practiced best to look up specific information or if the values must be reported precisely. However, figures and graph are best for illustrating trends and making comparisons therefore we used figures to provide reader friendly and easy to understand the trend and comparison.

Reviewer 2 Report

Ahmed et al. did great work and studied the role of glycine betaine in root morphology and responses of wheat seedlings under drought stress.

I have several comments to improve this manuscript.

  • The introduction is written very well.
  • Line 205: Please add more details about selected genes.
  • Add a section on materials and methods about statistical methods used to analyze data.
  • Please check the letters used in graphs to show the LSD result. For instance in Figure 2, the shoot fresh weight of 2-GB under water stress is higher than control from well water but "c" is used for control, and "d" is used for 2-GB under water stress. Please edit it.
  • Line 290: Please explain why these three hormones, ABA, SA, and JA, were chosen for the measurement.
  • Line 395: please use "(by measuring MDA)" instead (MDA).
  • Add it to line 396: Under abiotic stresses, uncontrolled free radicals are significantly increased and the concentration of antioxidants, nonenzymatic and enzymatic, is enhanced to regulate cellular homeostasis and reduce lipid peroxidation of the plasma membrane (Heidari et al, 2021).

Heidari et al, 2021: https://www.mdpi.com/2073-4395/11/6/1146

  • Line 434: delete the "." to link with the next sentence.

Author Response

We appreciate the reviewers’ comments and valuable suggestions. Followed by three Reviewers’ comments. We have revised the manuscript under the “track changes” mode. Our point-to-point responses are presented below.

Point1: I have several comments to improve this manuscript.

  • The introduction is written very well.
  • Line 205: Please add more details about selected genes.

Response: We would like to express our heartfelt appreciation to the reviewer for his or her constructive remarks. We provided supplementary table containing gene name and primer sequence. And we added ‘The supplementary Table 1 presents the gene-specific primers generated with DNAMAN software (Lynnon, Quebec, Canada)’ in the Line 206.

Point2: Add a section on materials and methods about statistical methods used to analyze data.

Response: We would like to offer our sincere gratitude to the reviewer for their helpful comments and ideas for enhancing our manuscript. The sub-section for Statistical analysis is added in material method section. (Line 211-215)

Point3: Please check the letters used in graphs to show the LSD result. For instance in Figure 2, the shoot fresh weight of 2-GB under water stress is higher than control from well water but "c" is used for control, and "d" is used for 2-GB under water stress. Please edit it.

Response: Thanks for this comment. We have revised the lettering in Figure 2.

Point4: Line 290: Please explain why these three hormones, ABA, SA, and JA, were chosen for the measurement.

Response: Thanks for this comment. Phytohormones play an important role in mediating plant responses to various abiotic stresses such as drought, salinity, cold and heat stress. Traditionally, Abscisic acid (ABA), salicylic acid (SA) and jasmonic acid (JA) have been, are associated with plant stress tolerance to different biotic and abiotic stress, therefore we selected these hormones the study the effect of GB root priming on hormonal homeostasis of these stress related hormones.

Point5: Line 395: please use "(by measuring MDA)" instead (MDA).

Response: Thanks. We have revised this manuscript as you suggested and at Line 395, the sentence is rearranged and used “by measuring MDA content”

Point6: Add it to line 396: Under abiotic stresses, uncontrolled free radicals are significantly increased and the concentration of antioxidants, nonenzymatic and enzymatic, is enhanced to regulate cellular homeostasis and reduce lipid peroxidation of the plasma membrane (Heidari et al, 2021).

Response: Thanks. We have revised the manuscript and above given paragraph is added in suggested position (line 396) with reference.

Point7: Line 434: delete the "." to link with the next sentence.

Response: Thanks. We have revised our manuscript and changed in Line 434, "." is deleted and the sentence is linked.

Reviewer 3 Report

This manuscript presents a well-organized research on Glycine betaine uses, its beneficial role  on wheat plant under drought condition. 

Regarding the organization of the manuscript, it contains a high number of bibliographic references and maybe for this reason no enough focus has been dedicated to each one of these.

All over the manuscript many different concepts are presented but not well linked each other. Many times, result difficult to understand the purpose of a sentence in the context of the paragraph. Also, an extensive English revision is needed; probably some terms wrongly used can add confusion to the reader.

Line 92 indicates that research was done 2year earlier, does this type of work not published yet.

Line 92 please indicate source of Jimai-22 variety of wheat

Line 104 Drought stress (WS) is WS correct for Drought stress, please check

Line 114 correct spelling of choosen

Line 133 300 μl of 0.750μm Nitroblue add comma before 300

Other comments include:

  • avoid repetition of keywords that are already in the article title.
  • Improve the quality of all Figure 
  • Decrease Plagarism percentage 
  • Furthermore, several sentences have misplaced commas that break the ideas in an unnatural way that makes comprehension difficult at times.

Author Response

We appreciate the reviewers’ comments and valuable suggestions. Followed by three Reviewers’ comments. We have revised the manuscript under the “track changes” mode. Our point-to-point responses are presented below.

Point1: All over the manuscript many different concepts are presented but not well linked each other. Many times, result difficult to understand the purpose of a sentence in the context of the paragraph. Also, an extensive English revision is needed; probably some terms wrongly used can add confusion to the reader.

Response: We would like to express our profound gratitude to the reviewer for your insightful comments and suggestions on how to improve our manuscript. We carefully revised the English expression, wrong use commas, and other editing mistakes to make our manuscript more clearly to readers.

Point2: Line 92 indicates that research was done 2year earlier, does this type of work not published yet.

Response: Due to pandemic and personal engagement the author was busy in teaching practices, project completion and lab development therefore delayed the process of manuscript finalization and publication.

Point3: Line 92 please indicate source of Jimai-22 variety of wheat

Response: thanks. The seeds of Jimai-22 wheat variety were bred and provided by the Shandong Academy of Agricultural Sciences. This information was added in Line 95.

Point4: Line 104 Drought stress (WS) is WS correct for Drought stress, please check

Response: Thanks. Typing mistake at line no: 104 is corrected and “Drought” is replaced with “Water”

Point5: Line 114 correct spelling of chosen

Response: thanks.   The correct past participle form from verb choose is chosen, therefore it will be more perfect with word “chosen” at line no:119.

Point6: Line 133, 300 μl of 0.750μm Nitroblue add comma before 300

Response: At line no: 133 “,” is added before 300 μl

Point7: avoid repetition of keywords that are already in the article title.

Response: Thanks for reviewer insightful comments and suggestions, the keywords are revised and repeating keywords are removed.

Point8: Improve the quality of all Figure 

Response: Thanks. All the figures are revised, and quality of figures are improved in Adobe photoshop

Point9: Decrease Plagiarism percentage 

Response: Thank you for your keen review, we carefully reviewed our manuscript and reduced plagiarism percentage.

Point10: Furthermore, several sentences have misplaced commas that break the ideas in an unnatural way that makes comprehension difficult at times.

Response: Thanks for the careful review of our manuscript. You mentioned a very critical point. According to your suggestion, we have carefully rechecked the punctuation and the meaningfulness of the sentences and improved it accordingly. Such as follows:

Line 144-145 the commas were deleted and the sentence was revised to ‘The lipid peroxidation in roots was assessed by the thiobarbituric acid (TBA) test which indicates MDA as a final product of lipid peroxidation.’,

Line 155, the comma after H2O2 was deleted.

All the commas used as thousand separators were deleted in Line 150 and 158 to make all expression consistently in the whole manuscript.

Line 185, we deleted the comma before ‘and’. 

Line 227, we deleted the comma after ‘Whereas’.

Line 297, the comma after ‘SA’ was deleted.
